# The Effects of an Eight-Week Integrated Functional Core and Plyometric Training Program on Young Rhythmic Gymnasts’ Explosive Strength

**DOI:** 10.3390/ijerph20021041

**Published:** 2023-01-06

**Authors:** Cristina Cabrejas, Mónica Solana-Tramunt, Jose Morales, Ainhoa Nieto, Ana Bofill, Eduardo Carballeira, Emanuela Pierantozzi

**Affiliations:** 1Department of Sports Sciences, Ramon Llull University, FPCEE Blanquerna, 08022 Barcelona, Spain; 2Royal Spanish Swimming Federation, 28007 Madrid, Spain; 3Department of Medical Sciences, Faculty of Medicine, University of Girona, 17071 Girona, Spain; 4Department of Physical Education and Sport, Faculty of Sport Science and Physical Education, Campus Bastiagueiro, University of A Coruña, 15179 Oleiros, Spain; 5Department of Neuroscience, Rehabilitation, Ophthalmology, Genetics, Maternal and Child Health, University of Genoa, 1700 Genoa, Italy

**Keywords:** rhythmic gymnastics, lower limb power, jumping ability, specific core strength, plyometric training, integrated training, rhythmic gymnastics performance

## Abstract

Background: Explosive strength is essential for rhythmic gymnasts’ performance. It has been suggested that core stability (CS) and plyometric training can enhance athletes’ explosive strength. Nevertheless, there is some uncertainty about the effects of integrated core and plyometric training (CPT) programs on rhythmic gymnastics (RG) performances. Purpose: to evaluate the effects of an integrated functional CPT program on young rhythmic gymnasts’ explosive strength and jump/leap performance. Method: We recruited 44 young (age = 10.5 ± 1.8 years old; peak height velocity, PHV = 12.2 ± 0.6 years old) female rhythmic gymnasts and randomly allocated them into a control group (CG) and an experimental group (EG). Pre and post-intervention, the explosive strength of both groups was assessed using countermovement jump (CMJ) and single-leg CMJ (SLCMJ) tests, conducted using a force platform, and expert RG judges evaluated their performance of RG-specific jumps. Before the post-test, the EG (*n* = 23) completed an 8 week functional CPT program based on RG technical requirements. Meanwhile, the participants in the CG (*n* = 21) received their regular training sessions. Linear mixed model analyses were applied to evaluate the effects of an intra-subject factor (TIME: pre-post) and an inter-subject factor (GROUP: control-experimental) on each dependent variable. When no significant interaction effect was found, Cohen’s d effect size was calculated. Results: After 8 weeks, the EG obtained significantly better results in all variables measured by the CMJ and SLCMJ (*p* < 0.01) tests. The judges’ scores indicated greater improvements in the EG after the CPT program in the stag and the split leap. Conclusions: An integrated functional CPT program improved explosive strength in a group of young rhythmic gymnasts and had a large impact on aspects of RG-specific performance. Coaches should consider using this CPT to improve RG performance.

## 1. Introduction

Strength is among the main physical capacities that have been found to contribute to performance in rhythmic gymnastics (RG) [1,2,3,4], and it is required for most of the movements and elements in this sport [5]. Successful technical performance, with the necessary range and intensity, is only possible with well-developed strength [6]. Therefore, strength is unquestionably a determining factor in high-performance RG [7].

The specific manifestations of strength in RG are strength-endurance and explosive strength [8,9]. There are two studies that compared the explosive strength of non-elite gymnasts to those of elite gymnasts [10,11], finding that the latter were more explosive jumpers. Gymnasts’ lower limbs must exert force explosively to execute RG jump/leap skills [12,13]. RG demands a good deal of vertical jumping ability [14]. Leaping effectively is an essential component of RG performance since gymnasts must execute complex leaps accompanied by choreographic elements [15]. Analyses of RG competition routines have pointed to a predominance of body difficulties in the jump/leap group [1]. This would suggest that jumping capacities are crucial for gymnastics. In RG, jumps are the main body element in training and competition routines [7].

RG judges observe and qualitatively assess jumps/leaps to determine their validity [16]. The FIG Code of Points [17] specifies that the successful completion of jump difficulties is characterized by a defined and fixed shape during flight and sufficient height (elevation) to show the corresponding body figure. This group of elements is assessed in terms of these two main aspects, referred to as range (elevation capacity) and shape (physical segment positioning during the phases of push-off, flight, and landing) [5]. The flight phase depends on the quality of the push-off, and the flight duration depends directly on the jump height [7]. The higher the athletes go during the jump, the greater their chances of succeeding in setting the body figure and, therefore, the greater the likelihood of receiving good scores from the judges [9].

In RG movements, isometric force plays a critical role in the pre-activation of the muscle before a contraction of the stretch-shortening cycle (SSC) [18]. Isometric strength exercises, such as those used in core stability (CS) exercises, are an innovative and highly effective ways of increasing muscle mass, toning, and developing strength levels to improve fitness and athletic performance [19,20]. The spine provides essential stability to the body, support for body weight, and, most importantly, cushioning for jumps [21]. In addition, the acceleration or deceleration of body segments in athletic performance depends upon the stability and strength of the core muscles that regulate the upper and lower extremities [22]. Thus, the core muscles are considered the kinetic link between the lower and upper limbs, and they are essential in effectively transferring force throughout the body [23,24]. Efficient control of the core muscles ensures efficient acceleration, deceleration, and stabilization [25] because all movements originate in, or are conducted through, the trunk [26]. Therefore, a CS program could improve athletes’ explosive strength.

Plyometric training (PT) improves athletes’ explosive strength and jumping ability [21,27]. PT is intended to enhance muscle force and power production through the SSC of the muscle unit [28] to maximize power production [29]. The SSC is characterized by a rapid transition between the initial eccentric “stretch” and subsequent concentric “recoil” [30]. Typically, PT involves exercises such as vertical jumps and box jumps that demand slower SSC or, on the other hand, bounding, repeated hurdle hops, and drop jumps, which call for a faster SSC [31]. Although previous studies have suggested that PT is not suitable for young people [32], more recent studies have argued that it is a safe and effective method to improve physical abilities such as vertical jump performance, rebound jump height, rate of force development (RFD), and it can help reduce leg stiffness in young athletes [33,34,35]. A study conducted with 17 to 20-year-old international rhythmic gymnasts concluded that a 4 week PT program improved their explosive strength [36]. Elsewhere, it has been reported that a year of PT improved the explosive strength of the lower limbs of adult rhythmic gymnasts [37]. A more recent study gave 6 weeks of PT to young rhythmic gymnasts, and the authors found an improvement in jump performance in the experimental group [38]. Another study evaluated RG leap ability after a month of training combining Pilates and water resistance plyometrics [14]. The authors concluded that the rhythmic gymnasts in their study improved their leaping ability right after the training intervention and maintained their improved performance upon follow-up measurements (three months and then one year later) [14].

In order to reach the top level of RG performance, it is fundamental to begin intensive training at a very young age, as this is the ideal time to start to learn and perform complex gymnastics skills [39]. RG training plans should be aimed both at the overall development of motor performance and, more specifically, at enhancing strength, jumping ability, and flexibility of the upper and lower limbs [1]. In girls, motor performance increases until the age of 13 or 14, but after that, there is only slight improvement [40]. It has also been reported that strength increases linearly until the age of 15, although there is less evidence of any adolescent spurt [40]. Thus, it is crucial to implement explosive strength training in later childhood and adolescence and to assess biological maturation when training young athletes to increase their strength.

Some necessary RG attributes can be developed simply by practicing regular gymnastics routines. However, PT seems likely to facilitate the further progression of athletes who have already reached a certain level of force and speed [41]. RG coaches often employ PT to improve the explosive strength of their gymnasts, and it is frequently part of the daily training routine [42]. Therefore, special strength training is viewed as necessary and must meet the sport’s specific demands, in this case, by improving gymnastics skills [43]. Nonetheless, strength conditioning remains the most neglected area of RG preparation [13]. That is why a specific CS and PT program has the potential to benefit RG performance in the early stages.

CS training protocols improve explosive strength and jump performance in RG and artistic gymnastics [44,45]. It is suggested that PT and CS programs could enhance explosive strength and RG jump performance. Nevertheless, to the best of our knowledge, the effects of a combined functional CS and PT program on young rhythmic gymnasts’ explosive strength have yet to be tested. Therefore, this study aims to evaluate the impact of integrated functional core and plyometric training (CPT) on the explosive strength of young rhythmic gymnasts and on their performance of specific RG jumps. 

## 2. Materials and Methods

### 2.1. Design

We carried out a randomized parallel clinical trial. The study was conducted according to the CONSORT standards [46].

### 2.2. Participants

We used GPOWER v3.1 software (Bonn FRG, University of Bonn, Department of Psychology) to make an a priori calculation of the sample size needed to obtain a Power (1 − ß) > 0.9, effect size = 0.4, and α = 0.05. The total required sample was 36 participants. In the end, we screened 59 participants for eligibility. However, only 45 participants met the inclusion criteria (Figure 1), and we decided to recruit all of them to ensure that we had sufficient participants in case of sample attrition. Ultimately, only 44 participants were analyzed because one participant in the control group did not complete the post-test.

All the participants gave their consent to participate in the study, and the procedures followed the latest version of the WMA Declaration of Helsinki. The gymnasts recruited must meet the three inclusion criteria: (1) training regularly at least three times a week, (2) having trained in gymnastics for more than a year without long breaks, and (3) having competed in regional federated and school RG competitions (Figure 1). Participants were excluded if they had any pain or injury that could disturb the training sessions or the testing. Participants were randomly assigned either to the control group (CG, *n* = 22) or the experimental group (EG, *n* = 23) using the online randomization software Research Randomizer (randomizer.org) (Figure 1). Gymnasts were asked not to consume any medicine or drinks that could disturb the nervous system. Descriptive characteristics are shown in Table 1.

The ethics committee of Ramon Llull University in Barcelona approved this study (CER- FPCEE Blanquerna, 1819007D). Parents or legal guardians of the participants completed the informed consent document prior to the study.

### 2.3. Procedures

The EG underwent the 8 week combined functional CPT program, featuring three 30 min sessions a week (a total of 24 sessions), from 19 March 2020 until 15 May 2020. The CG followed their regular training regimen over the same number of sessions.

All participants were tested 2 weeks before and2 weeks after the 8 week intervention. Prior to the testing session, the order of the participants and the two tests was randomly determined using a true random number generator to control for bias. Jump tests were carried out over 2 weeks, in the pre-test and post-test, by a sport-specialized physiotherapist unaware of each subject’s allocation group. The gymnasts performed a 15 min warm-up of cardiovascular activation and dynamic stretching exercises before the tests. 

### 2.4. Training Protocol

The present study was registered on clinicaltrials.gov with the ID number NCT04663633. The program involved exercises challenging the gymnasts’ core stability and postural control, and it included SSC explosive strength exercises executed with specific RG elements and postures (Table 2). The CPT program was designed following the training guidelines suggested by Bompa [21,47], who found trunk strength to be a key element in strength training in childhood and adolescence. In the plyometrics part of each training session, it is desirable to incorporate low-impact plyometric exercises (hops, jumps with two and one legs, and jumps from benches at the height of 25 and 30 cm) to acquire future adaptations and prevent injuries in children and adolescents [48]. Previous studies have shown that rhythmic gymnasts perform qualitatively and quantitatively better when trained with methods other than RG traditional training [37].

Participants in the EG were encouraged to hollow their navels while exhaling during each repetition of the CS exercises. They had to perform each repetition simultaneously with the breathing cycle, and they kept their lumbopelvic area neutral and stable while performing RG specific jumps and balance elements. 

The functional integrated CPT program contained three blocks. The first block consisted of a specific CS and plyometric skill circuit using cones, hurdles, and unstable surfaces. The second block included plyometric exercises involving RG jumps and balances combined with CS actions. Finally, the third block consisted of CS exercises in prone, supine, and side positions on the floor, combining specific balance poses that place demands on the core, as well as standing-up balances, with a series of jumps. We selected three different leaps (scissors, stag from assemble, and split leap) and three balances (passé, side with help, and arabesque) all of which are very common in novice-to-intermediate RG. These specific exercises are intended to convert the integrated CPT into sport-specific functional training, and they were selected to encompass various lower limb position planes and different techniques. It is advisable to employ exercises with the potential to challenge the core musculature, in all three planes and ranges of movement, to fully develop CS [49]. All RG exercises were executed balanced on both the right and left sides (exercises shown in Appendix A). Participants performed core and balance exercises on unstable surfaces and on softballs to stimulate anticipatory adjustments of the stabilizing muscles when they were trying to minimize postural destabilization [50]. An expert CS physiotherapist and a professional RG coach, both members of the research team, developed the training protocol, and the gymnasts underwent the CPT program under the supervision of professional RG coaches.

Every session, participants rated their internal perceived intensity in the rate of perceived exertion session scale (sRPE). The sRPE is a valid method of quantifying the effects of a training session through the assessment of the internal training load that can be applied in a wide variety of exercises [51]. A CPT trial was applied to determine the gymnasts’ perceived exercise intensity. The training load to achieve a prescribed number of repetitions was adjusted to 7–8 values in the RPE scale (i.e., hard). Gymnasts from both groups scored on the perceived exertion scale 30 min after every CPT session [51]. The score provided was multiplied by the minutes performed during the training session to obtain the sRPE [52]. There were no significant differences in sRPE between groups. This outcome was used to modulate the CPT training periodization plan. When values were lower than 7–8 sRPE, the load was augmented, incrementing the maintenance time of correctly executed isometric exercises and adding a set in every exercise that was technically well-executed.

During sessions, the EG and the CG did their general warm-up simultaneously, which consisted of approximately 30 min of general activation and stretching exercises. After that, the EG did the CPT program, while the CG completed their regular conventional RG-specific warm-up, combining traditional flexibility, strength, and abdominal exercises, all aimed at RG body techniques (splits, bridges, leg kicks, body waves, v abs, candlesticks, feet work, etc.) (detailed RG warm-up exercises and training load are shown in Appendix A). Even though the CG also performed core exercises during the RG traditional warm-up, all the CPT exercises, without exception, contained a high demand for core stabilization, implying highly demanded isometric strength and control of the core muscles. Since SSC exercises were not present in the CG RG warm-up, the EG received more specific loads of CS and SSC explosive strength exercises than the CG. Afterward, the two groups reunited to undergo the rest of their regular training (the weekly plan is shown in Appendix A).

The integrated functional CPT protocol was included in the competitive macrocycle (second half of the season) with a frequency of three times per week, as suggested previously [37].

### 2.5. Measurements

CMJ and single-leg CMJ (SLCMJ) tests were used to assess the explosive strength of rhythmic gymnasts. RG jumps use either one-leg or two-leg push-off techniques, which justifies the selection of bilateral (CMJ) and unilateral (SLCMJ) jump tests. The two tests were administered in a random order to each of the gymnasts.

### 2.6. Testing Procedures and Instrumentation

#### 2.6.1. Counter-Movement Jump (CMJ) Test

The participants performed the CMJ from a standing position in accordance with Bosco’s [53] protocol. The gymnasts stood on the board barefoot and placed their hands on their hips to eliminate the influence of the potential impulse to move their arms. Participants were instructed to jump as fast and high as possible, first lowering their center of gravity and adopting a half squat position (i.e., descend the knees to 90°, feet shoulder-width apart) and, then, without a pause, jumping, pushing vertically off the ground. They had to land with their ankles extended (toes pointed). Each subject completed three jumps, with 1 min of rest between them, and the jump with the highest power value was used for further analysis (Figure 2). 

#### 2.6.2. Single Leg Countermovement Jump (SLCMJ) Test

The participants performed SLCMJ from a standing position with one bare foot on the board and the free lower limb bent and not touching the force plate. The gymnasts were instructed to place their hands on their hips and jump as fast and high as possible. The first movement consisted of a single-leg half squat, i.e., knee at ~90° and, then, without a pause, a single-leg jump pushing vertically off the ground. Each participant completed six jumps (three with the right leg and three with the left as support) with 1 min rest periods between them. The jumps with the highest power value for the right and the left lower limbs were used for further analysis (Figure 2).

Data for the CMJ and SLCMJ tests were collected using a force plate (Kistler 9260AA, Winterthur, Switzerland) connected to a data acquisition system (Kistler 5695b, Winterthur, Switzerland). We employed the MARS software (Kistler, Winterthur, Switzerland) to acquire and store the raw data (sampling rate 1000 Hz) and to calculate all dependent variables. Calibration of the system was performed according to the MARS software recommendations. 

The CMJ and SLCMJ variables measured were: jump height of flight (Height), which was the height of the jump calculated from the deformation of the platform’s force sensors, measured in meters; this parameter was calculated from the take-off speed. Vertical take-off velocity (Take-off) was the velocity of the vertical movement at the time of take-off, calculated from flight time, measured in m/s. The average power (AVE P) is measured in watts (W). Moreover, the maximum concentric rate of force development (RFD)—P3 was also calculated. P3 is the software designation for the portion of the force curve used to calculate the RFD. The force platform software (MARS) calculates the RFD from the maximum slope of the force curve from the start of the concentric movement to the force peak.

The RFD parameter indicates maximal force development in a unit of time and is an index of explosive strength [54]. RFD is an essential contributor to vertical jump performance [55].

#### 2.6.3. Rhythmic-Gymnastics-Specific Jump Tests

The participants performed three RG jumps—the stag leap, the scissors leap, and the split leap (Figure 3). The gymnasts were allowed to perform a chassé before pushing off to jump (as commonly occurs in competition). Each gymnast performed three trials of each specific RG jump, with 1 min of rest between trials, and the order of the jumps was randomly determined. There were three national and international RG judges who evaluated the technical execution of each RG jump. All trials were recorded by a video camera (Sony, HDR-CX625, China) attached to a tripod, and the judges subsequently assessed the recorded performances. The jumps’ execution was evaluated according to the guidelines of the F.I.G. Code of Points [56]. The judges issued a final score after subtracting points for execution deductions and technical faults. Minor faults led to deductions of 0.10 points; medium faults came with a deduction from 0.20 to 0.30 points; large faults lowered the final score by up to 0.50 points or more. Judges considered the dimensions of body placement, amplitude, elevation, landing, and cleanliness. Poor jumping execution resulted in greater deductions, while well-performed jumps led to lower deductions. The performance with the best score for each trial and gymnast was recorded for further analysis, and the final score was the average of the scores from the three judges. 

#### 2.6.4. PHV Age

Strength is affected by age [40]. It is suggested that children increase their strength linearly until the age of 15 [40]. The present study recruited both prepubescent and pubescent rhythmic gymnasts whose ages ranged from 8 to 15 years. Therefore, the participants’ varying degrees of maturation must be considered. We estimated the age at PHV with a multiple regression equation that used anthropometric measures of standing height, sitting height, leg length, and weight [57]. This equation estimated the time interval between the predicted age at PHV and the individual’s actual age. The obtained values could be negative, which indicated that the PHV age had not yet been reached. If the values were positive, the result would be interpreted as the age of PHV surpassed. Finally, if obtained values were zero (0), it was interpreted as the current age being the exact age of PHV [57]. An ISAK-certificated researcher carried out all anthropometric measurements according to standard ISAK procedures.

### 2.7. Statistical Analyses

The descriptive data on the variables were presented as mean ± SD. The dependent variables included Height, Take-off, Average Power, and RFD, as measured in the CMJ and the SLCMJ with right and left leg tests. The other dependent variables were the scores awarded by the expert judges for the performance of scissors, stag jump, and split leap under different conditions.

Linear mixed models for repeated measure designs were employed to analyze the changes and differences between groups when all assumptions were met. A normality analysis of the residuals was carried out using the Kolmogorov–Smirnov test for each variable, and no deviations from a normal distribution were found. Homoscedasticity was examined by plotting the residuals-predicted value. This analysis indicated that the residuals were constant for all the predicted values. An alpha level of *p* < 0.05 was established for all the analyses.

GAMLj was used to carry out the linear mixed model analyses. This module relies on the R formulation of random effects. Specifically, it was done using the lme4 R package in Jamovi software (https://www.jamovi.org/ accessed on 21 July 2022). GAMLj estimates variance components with restricted (residual) maximum likelihood, which, unlike earlier approaches to estimating maximum likelihood, yields unbiased estimates of variance and covariance parameters. The fixed effects established in the analysis were the inter-subject group factor (EG or CG), the intrasubject time factor (PRE and POST tests), and the interaction (GROUP × TIME). The random effect included the participants’ intercepts. The effect of the intervention was estimated by the β coefficient and its 95% confidence interval (CI), representing a non-standardized effect size. 

Concomitantly, when no significant interaction effect was found, Cohen’s d effect size was calculated on all scores awarded by the expert judges (scissors, stag jump, and split leap), in the pre- and post-tests, to measure small (0.2), moderate (0.5), and large (0.8) effect sizes [58].

## 3. Results

The results showed no differences between the study groups before the intervention regarding the variables analyzed.

Table 3 presents the effects of the fixed factors, obtained after analyzing the CMJ and SLCMJ variables, using the linear mixed model. A significant interaction effect was found in all situations. 

Table 4 shows the descriptive results and Post Hoc within subject comparisons of the CMJ and SLCMJ variables.

The results, in terms of the scores for the RG leaps (stag, scissors, and split), are detailed below. The linear mixed model analysis, applied to the three RG jumps’ scores from the RG judges, showed no significant interaction between the factor time × group (Table 5). The same analysis showed a significant effect of the time factor on the scores for the three exercises (stag, scissors, and split). The pairwise comparison showed a significant improvement (*p* < 0.001) of the EG and CG after the intervention in scores for the stag, the scissors, and the split leap, indicating that all scores were better in the post-test (Figure 4).

Effect size results indicated a large effect on the stag and the split leap scores in the EG, as well as a medium effect on those of the CG. A large effect size is found in the scissor leap for both groups. After the CPT program, the EG recorded better results than the CG in overall RG leaps (Figure 5). 

## 4. Discussion

This study aimed to analyze the effectiveness of an 8 week RG-based CPT program on the explosive strength of young rhythmic gymnasts. As far as we know, this is the first study to evaluate the effects of a CPT program on the explosive strength of young rhythmic gymnasts. The main findings comprise significant improvements in the EG in CMJ, as well as right and left SLCMJ, post-test results in all the dependent variables (Height, Take-off, AVE P, and RFD). The judges’ scores for the RG-specific jump tests did not indicate significant differences between the EG and the CG in any of the three leaps evaluated (stag, scissors, or split). Indeed, at the time of the post-test, both groups significantly improved their scores on the RG-specific leaps. However, it is worth noting that an examination of the particular kinds of jump shows large effect sizes on the stag and the split leap in the EG, while only a medium improvement in the CG appears in the post-test. Both groups displayed a large effect size in their post-test scissors leap scores. The results of the pre-tests did not show any significant differences between the EG and CG, so the differences obtained in the post-test can be attributed to the functional integrated CPT. 

These results suggest that adding an integrated functional CPT program to regular RG training can provide young rhythmic gymnasts with greater explosive strength than they would achieve if performing only RG-specific training. To the best of our knowledge, no prior study has evaluated young rhythmic gymnasts’ explosive strength after completing a CPT training program. However, a similar study involving integrated training was found. After 1 month of integrated training using Pilates and water resistance plyometrics, improved results elite rhythmic gymnasts’ jump height and explosive strength were found [14]. This training program is similar to ours in that it brings together core exercises (Pilates training) and plyometrics, but in the previous study, the exercise was not RG-specific, the training duration was shorter, and the participants were elite gymnasts instead of young competitive gymnasts. 

Some earlier studies have featured PT programs and evaluated rhythmic gymnasts’ explosive strength. For example, Taktak et al. [36] reported that the explosive strength of adult rhythmic gymnasts, as measured by a CMJ test, improved after a 4 week PT. In another research paper detailing a PT program, after 12 months during which a group of gymnasts added plyometric exercises to their training routine, they performed better than another group trained with regular exercises, with the PT group displaying increased agility, as well as greater strength in the lower limbs, on the vertical jump and horizontal jump tests [37]. In the most recent study, young rhythmic gymnasts were evaluated after completing 6 weeks of PT, and their jump performance improved [38]. Although the findings of all these studies are consistent with our results, the training programs used were traditional PT and did not include any specific RG core or plyometric exercises, nor did they feature any specific RG testing procedures. Studies using PT programs were also found in artistic gymnastics. Marina and Jemni [59] analyzed the jump performance of nine female elite-oriented gymnasts. The authors confirmed that a combination of heavy strength training and high-impact plyometrics is effective for prepubescent gymnasts. In another study, Hall et al. [60] evaluated the effects of a 6 week PT intervention, added to regular gymnastics training, on handspring vault performance and lower body power development in young gymnasts. They presented a small, but significant, increase in post-vault time in the air. All this rhythmic and artistic gymnastics research suggests that a PT program can help improve gymnasts’ explosive strength. 

Meanwhile, core training might also have helped improve the gymnasts’ explosive strength. Since the core provides greater stability for the movements of the lower extremities, from proximal to distal, improving core function may enhance the legs’ power [23]. It is suggested that strong core muscles function as hubs in the biological motor chain, acting as a fulcrum for the four limbs’ strength and establishing a channel for the cohesion, transmission, and integration of the upper and lower limbs [61]. CS training could improve the ability of the nervous system to organize muscle coordination and enhance efficiency in sports [61]. Indeed, stronger core muscles transfer strength to the limbs economically and harmoniously, and they also help athletes to maintain the stability of their trunk and hip joints. These adaptations allow gymnasts to execute more coherent, coordinated, and stable complex technical movements in the air [61]. A thesis reported that a 6 week core training program improved the vertical jump performance of young artistic gymnasts [44]. Elsewhere, a study including 120 Bulgarian rhythmic gymnasts found a significant correlation between back hold and maximum vertical jump [13]. Reviewing the literature, we only found three works that studied the effects of CS training programs in a young rhythmic gymnast population. There were two studies mentioned earlier that showed an improvement in CS parameters and indicated that these improvements might enhance RG performance, although explosive strength was not measured [62,63]. The other RG study found significant improvements in the gymnasts’ power, balance, and endurance after the CS training [45]. These results are consistent with our findings; nevertheless, that study did not include plyometric and specific RG exercises in the training program. CS training programs have been conducted in basketball [64], improving explosive strength parameters of pre-pubescent and pubescent basketball players, and similar results were found in soccer [65]. Finally, a study comparing RG resistance training and regular resistance training provided further evidence that extensive repetition of low-load resistance exercises is advisable for young rhythmic gymnasts to improve power and stiffness, and specific training appears to be preferable to increase reactive strength [66].

The significant improvement of the EG in the post-test in the CMJ and SLCMJ right and left tests on all variables measured (Height, Take-off, AVE P, and RFD) suggest that our functional training helps enhance explosive strength in young rhythmic gymnasts, thus improving physical parameters of the gymnasts’ vertical jump. The jump height variable is related to the length of flight time during the jump, the vertical take-off with the speed and coordination of this phase of the leap, and the power and RFD variables of explosive strength indicators [54,55]. Particularly, the evaluation of RFD during rapid contractions has recently become quite a popular method of assessing athletes’ explosive strength [67]. RFD seems to be mainly determined by the capacity to produce maximal voluntary activation in the early phase of an explosive contraction (first 50–75 ms), mostly due to an increased motor unit discharge rate [67]. 

These results also suggest that the CPT program helped gymnasts improve their performance throughout the three jump phases (take-off, flight, and landing). A prior study of split leap performance in RG highlighted the importance of the take-off and flight phase total time [68]. These temporal kinematic variables have been associated with better execution of this body difficulty. According to Hay [69], in the take-off phase, explosive strength in the lower limbs is required to generate the impulse action (which must be performed with great speed and coordination between the joints involved). It has been observed that the longer gymnasts take to perform the take-off phase, the worse jump performance they tend to display [68]. Moreover, the duration of the flight phase depends on take-off quality. The better the take-off, the greater the chances the gymnast will be able to perform specific actions and the greater the degree of difficulty that can be achieved with the jump [69]. Our results showed an improvement in the take-off parameters and flight time, both of which have been tied to the proper execution of RG leaps [68,70,71]. Furthermore, since the RG code of points establishes a series of requirements for the proper execution of RG jumps (“defined and fixed shape” and “height sufficient to show the corresponding shape”), flight phase duration takes on great importance. In other words, the results we obtained in this phase coincide with what is required by the code of points itself [17]. Thus, the longer gymnasts remain in flight, the longer they will have to show and maintain or adjust the shape of the jump (in our case, the stag, the scissors, and the split leap). With more time, technical quality increases, and jumps receive higher scores [68]. The landing phase aims to absorb the impact of the support and adopt and maintain a stable position [72]. Based on the results of Cabrejas et al. [73] it is suggested that young gymnasts improve postural control after a CPT program, which might help improve the landing phase of the jump and increase their ability to absorb the landing impact more efficiently thanks to improvements in the explosive strength of the lower limbs. The predictable and conscious stimulation of the muscle effectors imposed by CPT is an effective way to stimulate the cortical proprioception of the ankle, knees, hips, and lumbopelvic joints [74]. Thus, rhythmic gymnasts might better position their joints correctly and increase their explosive strength in the take-off, flight, and landing phases of RG jumps. In our study, the EG and the CG trained a similar number of hours, and the EG obtained higher performance after the intervention; thus, we agree with Marina and Jemni (2014) that gymnasts should devote more time to physical condition training designed to optimize their plyometric skills. This kind of training leads to better jump/leap execution and decreases the training time needed for technical routines.

Regarding the judges’ overall scores, we did not find any significant differences between groups after the CPT program. However, a significant improvement associated with the time factor was found, indicating that both the EG and the CG improved their RG leap scores at the post-test time. A large effect was found for the stag and split leap in the EG compared to a medium effect on the CG. These results show a tendency toward greater improvement in the EG than in the CG after the functional training. 

A possible explanation for the results, in terms of the judges’ scores, might be that, for young beginning gymnasts, the leap technique is complex, as these athletes may not have fully developed the execution factors involved in scissors, stag, and split leaps. The execution factors include the physical components of explosive strength, leg flexibility, and proper leap movement techniques [75]. The judges primarily evaluate the flight phase, during which strength, flexibility, and coordination are crucial [76]. Our gymnasts improved their explosive strength parameters, but they might still need more time to improve flexibility, coordination, and leap technique to achieve better RG leap scores from judges. 

As discussed above, it is clear that the angular flexibility of the joints, especially the amplitude of the hips and the extension of the ankle joints (with pointed toes), is an important factor influencing RG leap execution [68]. Greater maximum angles of ankle extension have been associated with lower penalties and, therefore, with better performance levels—a fact that underlines the importance of keeping one’s toes extended in gymnastic sports, especially in RG, in order to achieve the required aesthetic [17] and, consequently, better quality and scores. The maximum amplitude angle between the hips is also closely linked to improved jump execution [68]. This relationship may have its origin in 1) the importance of having the widest possible joint range of motion in the execution of RG-specific jump skills and 2), consequently, the relevance of flexibility during the active–ballistic manifestation of this kind of body difficulty. This hypothesis is supported by several authors and publications [68,70,71,76,77,78]. A study that evaluated four gymnastics jumps (vertical, vertical with 360° rotation of the body, stag ring, and straddle jump) with countermovement and upper limb movement, performed by junior gymnasts aged 13 to 15, concluded that gymnasts with low flexibility levels were unable to adopt the jump shape during the flight [16]. It is suggested that, during strength training, flexibility training should be increased accordingly [66] in order to ensure that gymnasts strike a good balance between strength and flexibility, which is a prerequisite for RG performance [79]. 

Our findings suggest a tendency toward improvements in RG jumps in the EG, especially in the stag and the split leap. These two jumps require the widest hip amplitude, indicating that the training program helped the gymnasts improve their legs’ opening capacity and specific explosive force. However, the CPT program was not aimed at improving the technical aspects of RG jumps nor enhancing flexibility, which is why it may not have been reflected as much in the judges’ scores.

Our functional CPT is creative and closely tied to the reality and functional needs of RG since it is linked to RG motor patterns. In our study, gymnasts trained using CS exercises that demanded SSC explosive strength, balance, and postural control. The challenges imposed on explosive strength, CS, and balance are close to the RG skills’ demands. This intention is supported by Lederman [80] and Boyle [81], who claimed that the more specific the training, the more it will transfer to a given sport. 

This study considered the importance of biological maturity for physical performance by calculating the gymnasts’ PHV age, and no significant differences between the randomly assigned groups were found. 

In addition, other studies affirm that regular gymnastics training helps to obtain better results in jump height, take-off speed, and maximum vertical speed of the center of mass, all of which are essential for the adequate performance of vertical jumps [82]. This might explain the significant improvement of the CG in the RG leaps post-test, with a medium effect size on the judges’ scores for the stag and split leap, as well as a large effect size on those for the scissors leap. It seems that the integrated functional CPT program challenges the gymnasts’ explosive strength more than the RG traditional training, but it did not change enough to significantly improve the judges’ scores for RG-specific leaps.

There are certain issues and limitations connected to this study’s design. Some researchers have questioned the use of general jumping ability tests to assess athletes in specific sports. For example, a study by Grande et al. [83] did not find a statistical relationship between the results of these general tests and gymnastics results, causing them to reconsider whether it is appropriate to assess these athletes with generic tests of jumping ability since they, subsequently, are not reflected in their sports performance. This observation is of great value because generic test batteries are frequently used to assess significant aspects of jumps, even when there is no accurate transfer to the jump that athletes perform in their usual routine. Furthermore, Di Cagno et al. [84] showed a similarity between elite and sub-elite gymnasts in CMJ performance, probably, because CMJ is not a standard leg strength test for rhythmic gymnasts. These athletes are more specialized in technical leaping performance, suggesting that unspecific RG tests might not be the most suitable tools to evaluate specific rhythmic gymnasts’ jump performances. Nevertheless, numerous studies have considered CMJ over a force platform as the gold standard to measure explosive strength in athletes [42,53]. Regarding the length of the training program, previous integrated CS and plyometric programs carried out with athletes reported results after interventions of 4–12 weeks [36,85]. We opted for an 8 week training program, as it is in the average CST length, and it has been suggested that the frequency and duration should be no less than twice a week over 4 weeks [61]. Future studies are warranted to analyze the effect of longer interventions. We hypothesize that more remarkable adaptations could be achieved after more extended training periods since motor behavior changes take longer to be affected by training [86]. It is also important to point out that participants’ ages ranged from 8 to 15 yo; thus, results should not be generalized for other age categories. One more limitation of the study is that the control and experimental groups are not balanced since they do not have precisely the same number of participants, but the statistical tests used are robust in these circumstances, and the results obtained are still valid. 

To our knowledge, only one prior study has analyzed the impact of an integrated CPT training program on rhythmic gymnasts [14], and one previous work has studied the CS training program’s effect on rhythmic gymnasts’ power [45]. Elsewhere, it has been suggested that CS training helps improve core performance, which might benefit RG performance [62,63,73]. Most available research has tried to demonstrate a relationship between PT and explosive strength performance in RG [36,37,38] and artistic gymnastics [59,60], while one study has assessed explosive strength after resistance training featuring strength and plyometrics exercises [66]. However, none of these earlier studies focused on integrated or functional training programs, nor did they present similarities with RG regarding exercise typology and requirements. It has been suggested that integrated functional CPT might enhance young rhythmic gymnasts’ explosive strength and may lead to a tendency to improve RG leap scores. Nevertheless, more research is needed to compare the effects of traditional CPT with those of specific RG-based CPT on explosive strength, to determine the long-term effects on rhythmic gymnasts, and to develop other valid and reliable RG-specific leap tests to assess different RG leaps and other RG skills.

The most important practical application of the present study is that RG functional training, which combines core and plyometric exercises executed with specific RG balance and leap postures, improves young rhythmic gymnasts’ explosive strength. Gymnasts can take advantage of including more specific content in their warm-up, and increasing their performance at the same time, if they try to be explosive in each jump and balanced in each jump reception; furthermore, they should exercise their core while keeping their lumbopelvic area straight and stable and hollowing the navel while exhaling. Considering these considerations, gymnasts will increase their performance and save time, reducing the necessity for extra sessions.

## 5. Conclusions

An 8 week RG-based CPT program enhanced explosive strength in young rhythmic gymnasts, specifically in terms of the variables of jump height, vertical take-off velocity, average power, and RFD measured during a CMJ, a right SLCMJ, and left SLCMJ tests using a force plate. The judges’ scores for RG leaps improved in both groups in the post-test, but there was a tendency toward the more remarkable improvement of the stag and the split leap scores in the EG. Hence, adding a functional CPT to regular training might help to improve rhythmic gymnasts’ explosive strength and RG jump performance-related variables. We encourage researchers to design studies with more extended training periods, analyzing the effects of CPT programs on the RG leaps and other RG key performance indicators.

## Figures and Tables

**Figure 1 ijerph-20-01041-f001:**
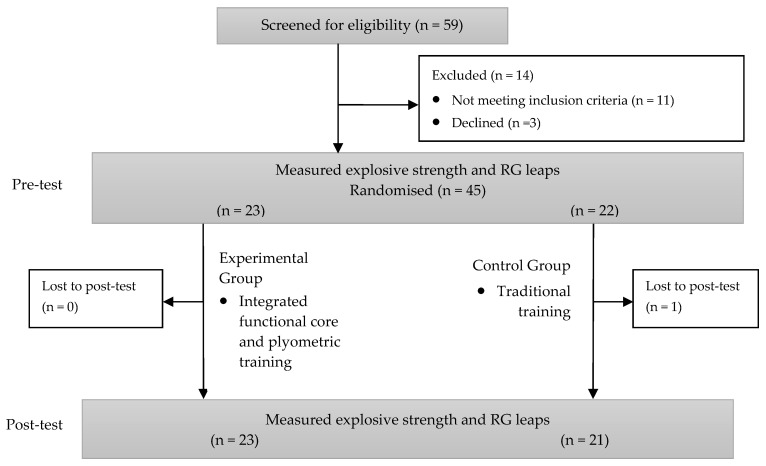
Sample flowchart.

**Figure 2 ijerph-20-01041-f002:**
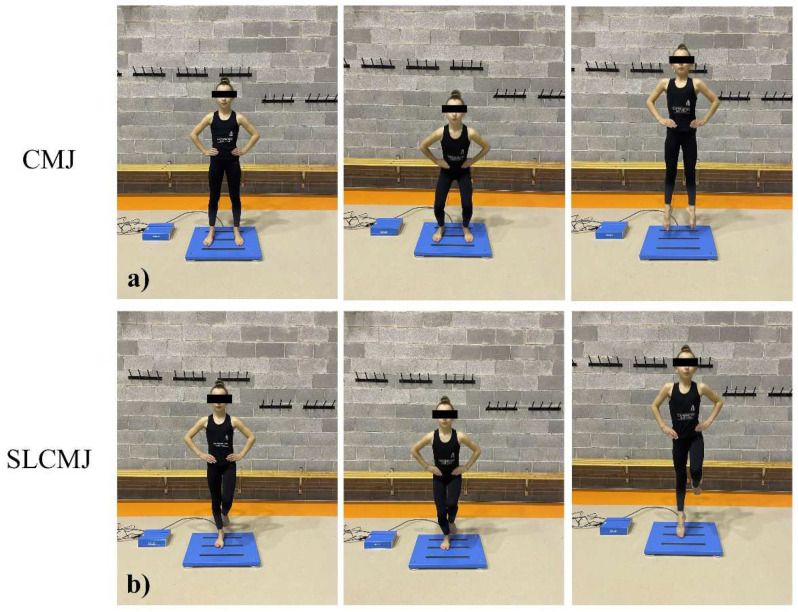
(**a**) CMJ test and (**b**) SLCMJ test. From a standing position, gymnasts performed an SSC fast movement to jump.

**Figure 3 ijerph-20-01041-f003:**
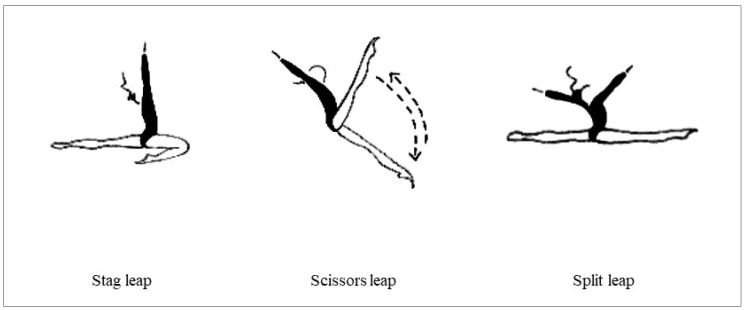
RG-specific jump tests performed [17].

**Figure 4 ijerph-20-01041-f004:**
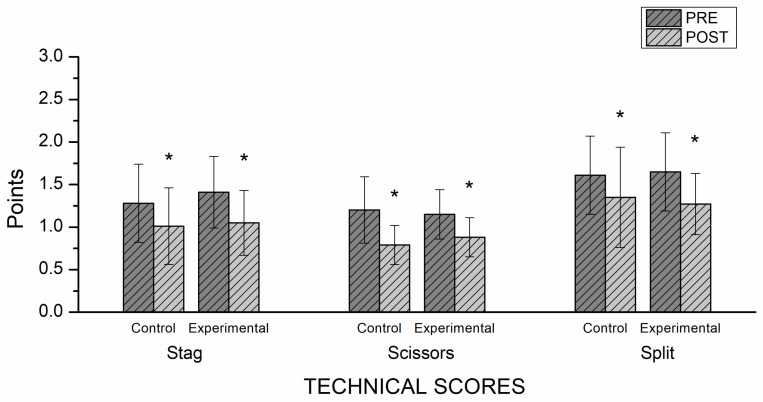
Judges’ technical score results. * Significant differences pre–post (*p* < 0.05).

**Figure 5 ijerph-20-01041-f005:**
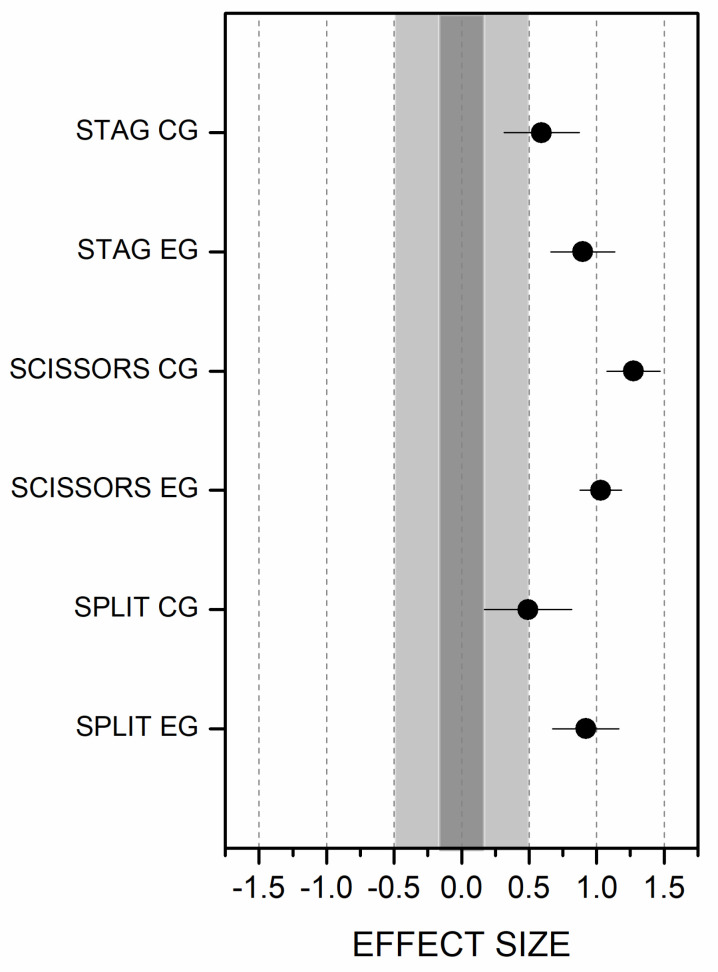
Standardized effect size of differences between pre and post-test RG leap scores.

**Table 1 ijerph-20-01041-t001:** Participants’ Characteristics.

Variables	EG (*n* = 23)	CG (*n* = 21)	T-Test (*p*-Value)
Age (years)	10.52 ± 1.90	10.43 ± 1.78	0.874
Peak height velocity (PHV) (years)	12.25 ± 0.55	12.23 ± 0.67	0.917
Years from PHV	1.21 ± 1.41	1.10 ± 1.38	0.798
Height (m)	1.44 ± 0.10	1.44 ± 0.11	0.801
Weight (kg)	37.82 ± 9.83	38.2 ± 8.03	0.892
Body mass index (kg/m^2^)	18.08 ± 2.56	18.06 ± 1.56	0.982
Height of center of gravity (m)	0.87 ± 0.07	0.87 ± 0.07	0.922

No significant differences between groups.

**Table 2 ijerph-20-01041-t002:** Overview of the 8 week integrated functional CPT program.

Exercises	W1		W2		W3		W4		W5		W6		W7		W8	
	Sets	Rep	Sets	Rep	Sets	Rep	Sets	Rep	Sets	Rep	Sets	Rep	Sets	Rep	Sets	Rep
Block 1																
Lateral hip bridge over bossu	2	6	2	6	3	6	3	6	3	6	3	6	4	6	4	6
Prone plank over the ball	3	6	3	6	3	6	3	6	3	6	3	6	4	6	4	6
Plyometric hops + RG balances	2	6	2	6	2	6	2	6	2	6	2	6	2	6	3	6
Plyometric double hops + RG balances	2	6	2	6	2	6	2	6	2	6	2	6	2	6	3	6
Plyometric RG jumps with cones	2	8	2	8	4	8	4	8	4	8	6	8	6	8	6	8
Plyometric RG jumps + hurdles	4	8	4	8	4	8	4	8	4	8	6	8	6	8	6	8
RG balances over balance disc	2	3 × 5”	2	3 × 5”	2	3 × 5”	2	3 × 5”	2	3 × 5”	2	3 × 10”	2	3 × 10”	2	3 × 10”
Block 2																
DJ + passé balance	1	2	1	2	1	2	1	2	1	2	1	2	1	2	1	2
DJ + arabesque balance	1	2	1	2	1	2	1	2	1	2	1	2	1	2	1	2
JDJ + passé balance	1	2	1	2	1	2	1	2	1	2	1	2	1	2	1	2
JDJ + arabesque balance	1	2	1	2	1	2	1	2	1	2	1	2	1	2	1	2
JDJ + side balance	0	0	0	0	1	2	1	2	1	2	1	2	1	2	1	2
DJ + stag jump	1	2	1	2	1	2	1	2	1	2	1	4	1	4	1	4
DJ + scissors jump	1	2	1	2	1	2	1	2	1	2	1	4	1	4	1	4
DJ + split leap	1	2	1	2	1	2	1	2	1	2	1	4	1	4	1	4
Block 3																
Lateral plank arm straight	2	6	2	6	2	6	2	6	2	6	4	6	4	6	4	6
Lateral plank with elbows	2	6	2	6	2	6	2	6	2	6	4	6	4	6	4	6
Hip bridge with passé balance	2	4 + 8”	2	4 + 8”	2	4 + 16”	2	4 + 16”	2	4 + 16”	2	4 + 16”	2	4 + 16”	2	4 + 16”
Hip bridge with side balance	2	4 + 8”	2	4 + 8”	2	4 + 16”	2	4 + 16”	2	4 + 16”	2	4 + 16”	2	4 + 16”	2	4 + 16”
Hip bridge with passé balance + bounces	2	4	2	4	2	8	2	8	2	8	2	8	2	10	2	10
Hip bridge with side balance + bounces	2	4	2	4	2	8	2	8	2	8	2	8	2	10	2	10
Passé balance + jumps	2	4	2	4	2	8	2	8	2	8	2	8	2	10	2	10
Arabesque balance + jumps	2	4	2	4	2	8	2	8	2	8	2	8	2	10	2	10
Side balance + jumps	2	4	2	4	2	8	2	8	2	8	2	8	2	10	2	10

Note. Reprinted with permission from “The Effect of Eight-Week Functional Core Training on Core Stability in Young Rhythmic Gymnasts: A Randomized Clinical Trial” by C. Cabrejas, 2022, Int. J. Environ. Res. Public Health, 19, 3509 p. 5, doi:10.3390/ijerph19063509. Copyright: © 2022 by the authors. Licensee MDPI. DJ = drop jump, JDJ = jump-drop-jump, W = week, Block 1. Circuit = mixed CS, balance, and plyometric exercises. Each week contained three CPT sessions. All exercises were performed with the right and left sides. All planks and lateral hip bridges were executed with passé, side leg, and arabesque positions, and they were maintained for 2 sec in each position. All DJ and JDJ were performed with a 30 cm bench.

**Table 3 ijerph-20-01041-t003:** Fixed effects of the CMJ and SLCMJ variables.

CMJ Variables	Effect	Estimate	Lower 95% CI	Upper 95% CI	T_42_	*p*
HEIGHT (cm)	(Intercept)	0.19	0.18	0.20	73.94	<0.001
	CG vs. EG	0.01	−0.004	0.03	1.06	0.143
	POST vs. PRE	−0.003	−0.009	0.002	−2.80	0.211
	CG vs. EG × POST vs. PRE	−0.02	−0.03	−0.01	3.95	<0.001
TAKE OFF (m/s)	(Intercept)	1.93	1.96	1.99	73.94	<0.001
	CG vs. EG	0.07	−0.04	0.19	1.06	0.246
	POST vs. PRE	−0.01	−0.04	0.01	−2.80	0.220
	CGvs. EG × POST vs. PRE	−0.09	−0.15	−0.03	3.95	0.002
AVE P (W)	(Intercept)	830.52	756.03	905.02	21.84	<0.001
	CG vs. EG	2.03	−146.96	151.03	0.02	0.979
	POST vs. PRE	−34.18	−73.96	5.59	−1.68	0.1
	CG vs. EG × POST vs. PRE	−127.22	−206.78	−47.67	−3.13	0.003
RFD (N/s)	(Intercept)	5826.39	5393.3	6259.53	26.36	<0.001
	CG vs. EG	92.16	−774.1	958.43	0.20	0.83
	POST vs. PRE	−327.62	−584.2	−71	−2,50	0 .01
	CG vs. EG × POST vs. PRE	−1000.67	−1513.9	−487.43	−3.82	<0.001
**Right SLCMJ** **Variables**	**Effect**	**Estimate**	**Lower 95% CI**	**Upper 95% CI**	**T_42_**	** *p* **
HEIGHT (cm)	(Intercept)	0.19	0.18	0.20	73.94	<0.001
	CG vs. EG	0.01	−0.004	0.03	1.06	0.143
	POST vs. PRE	−0.003	−0.009	0.002	−2.80	0.211
	CG vs. EG * POST vs. PRE	−0.02	−0.03	−0.01	3.95	<0.001
TAKE OFF (m/s)	(Intercept)	1.93	1.96	1.99	73.94	<0.001
	CG vs. EG	0.07	−0.04	0.19	1.06	0.246
	POST vs. PRE	−0.01	−0.04	0.01	−2.80	0.220
	CG vs. EG × POST vs. PRE	−0.09	−0.15	−0.03	3.95	0.002
AVE P (W)	(Intercept)	830.52	756.03	905.02	21.84	<0.001
	CG vs. EG	2.03	−146.96	151.03	0.02	0.979
	POST vs. PRE	−34.18	−73.96	5.59	−1.68	0.1
	CG vs. EG × POST vs. PRE	−127.22	−206.78	−47.67	−3.13	0.003
RFD (N/s)	(Intercept)	5826.39	5393.3	6259.53	26.36	<0.001
	CG vs. EG	92.16	−774.1	958.43	0.20	0.83
	POST vs. PRE	−327.62	−584.2	−71	−2,50	0.01
	CG vs. EG × POST vs. PRE	−1000.67	−1513.9	−487.43	−3.82	<0.001
**Left SLCMJ** **Variables**	**Effect**	**Estimate**	**Lower 95% CI**	**Upper 95% CI**	**T_42_**	** *p* **
HEIGHT (cm)	(Intercept)	0.083	0.076	0.089	26.92	<0.001
	CG vs. EG	0.002	−0.009	0.014	0.47	0.640
	POST vs. PRE	−0.008	−0.012	−0.004	−3.88	<0.001
	CG vs. EG × POST vs. PRE	−0.012	−0.020	−0.003	−2.89	0.006
TAKE OFF (m/s)	(Intercept)	1.24	1.190	1.301	47.591	<0.001
	CG vs. EG	0.043	−0.059	0.146	0.823	0.415
	POST vs. PRE	−0.049	−0.079	−0.019	−3.221	0.002
	CG vs. EG × POST vs. PRE	−0.100	−0.160	−0.039	−3.249	0.002
AVE P (W)	(Intercept)	465.64	421.94	509.35	20.882	<0.001
	CG vs. EG	27.68	−59.73	115.09	0.62	0.538
	POST vs. PRE	−34.81	−54.53	−15.09	−3.459	0.001
	CG vs. EG × POST vs. PRE	−58.33	−97.78	−18.89	−2.89	0.006
RFD (N/s)	(Intercept)	4056.80	3536.52	4577.05	15.28	<0.001
	CG vs. EG	176.41	−864.11	1216.90	0.33	0.741
	POST vs. PRE	−254.4	−446.01	−62.89	−2,60	0.013
	CG vs. EG × POST vs. PRE	−191.33	−574.40	191.84	−0.97	0.033

**Table 4 ijerph-20-01041-t004:** CMJ, right SLCMJ, and left SLCMJ descriptive results and Post Hoc comparisons within subjects.

Variables		Control Group				Experimental Group				*p*-Value
		Pre		Post		Pre		Post		
		Mean	SD	Mean	SD	Mean	SD	Mean	SD	
CMJ	HEIGHT (m)	0.19	0.03	0.19	0.03	0.19	0.03	0.21	0.03	0.006
	TAKE OFF(m/s)	1.91	0.23	1.88	0.22	1.93	0.19	2.11	0.2	0.014
	AVE P (w)	844.23	253.04	814.8	266.63	782.65	235.83	880.44	284.67	0.007
	RFD (N/s)	5866.67	1760.49	5693.95	1709.15	5458.49	1351.01	6286.45	1267.99	<0.01
Right SLCMJ	HEIGHT (m)	0.08	0.02	0.08	0.01	0.08	0.02	0.1	0.02	<0.01
	TAKE OFF(m/s)	1.23	0.19	1.22	0.14	1.25	0.13	1.34	0.16	<0.01
	AVE P (w)	490.94	172.93	503.12	168.15	496.24	149.47	559.2	145.19	<0.01
	RFD (N/s)	3863.38	1518.88	4007.62	1542.47	4010.85	1461.88	4617.26	1405.44	<0.01
Left SLCMJ	HEIGHT (m)	0.08	0.02	0.08	0.02	0.08	0.02	0.1	0.02	<0.01
	TAKE OFF(m/s)	1.23	0.2	1.23	0.19	1.22	0.16	1.32	0.18	<0.01
	AVE P (w)	448.98	158.76	454.62	149.75	447.49	145.81	521.47	151.81	<0.01
	RFD (N/s)	3889.19	1229.31	4048	1290.01	3969.94	1240.35	4320.03	1193.21	0.013

HEIGHT = jump height; TAKE OFF = vertical take-off; AVE P = average power; RFD = rate of force development. *p*-value within subjects= significant differences pre-post (*p* < 0.05). The *p*-value represents significant differences between EG and CG (*p* < 0.05).

**Table 5 ijerph-20-01041-t005:** Fixed effects of the technical scores’ variables.

Technical Scores Variables	Effect	Estimate	Lower 95% CI	Upper 95% CI	T_42_	*p*
STAG (points)	(Intercept)	1.185	1.075	1.294	21.16	<0.001
	CG vs. EG	0.087	−0.131	0.307	0.782	0.438
	POST vs. PRE	0.313	0.189	0.437	4.947	<0.001
	CG vs. EG × POST vs. PRE	0.083	−0.165	0.331	0.657	0.514
SCISSORS (points)	(Intercept)	1.004	0.932	1.077	27.075	<0.001
	CG vs. EG	0.021	−0.124	0.166	0.286	0.776
	POST vs. PRE	0.335	0.243	0.427	7.166	<0.001
	CG vs. EG × POST vs. PRE	−0.140	−0.324	0.042	−1.503	0.140
SPLIT (points)	(Intercept)	1.468	1.337	1.599	21.942	<0.001
	CG vs. EG	−0.024	−0.287	0.237	−0.185	0.854
	POST vs. PRE	0.319	0.223	0.415	6.517	<0.001
	CG vs. EG × POST vs. PRE	0.123	−0.068	0.315	1.261	0.214

## Data Availability

All data files are available from the figshare database: https://doi.org/10.6084/m9.figshare.21229505.v1, accessed on 30 December 2022.

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
