# Peer review of "The Effects of an Eight-Week Integrated Functional Core and Plyometric Training Program on Young Rhythmic Gymnasts’ Explosive Strength"

_ijerph, 2023, doi:10.3390/ijerph20021041_

Round 1

Reviewer 1 Report (Previous Reviewer 2)

Based on the comments, the authors significantly approved the manuscript. They provided extended changes mainly concerning the data analysis. Even the authors reduce the discussion part, from my point of view is still too long. However, as I mentioned before, this is a quality paper, and I recommend publishing it in IJERPH.

Author Response

[A] = answers from the authors.

Thank you again for your comments, all of them strengthen our work.

The discussion was hard to reduce since we think that all discussion information is important and it is hard for us to leave aside the details. Nevertheless, we will take your advice for further research and try from the beginning to write shorter discussions and avoid long writings.

Thank you, we appreciate your time and effort.

Reviewer 2 Report (Previous Reviewer 1)

I believe the authors have addressed all my concerns. However, I quibble with one of the statements in their response:

“Since the RPE results did not show significant differences between the two groups in terms of load and volume, the EG is not doing more than the CG.”

I do not believe that RPE is a replacement for measures of volume and intensity. RPE is used as a measure of intensity or more recently as a measure of “training load”, but traditionally, training load would be determined by volume x intensity. It is possible that the RPE is not different between groups, but the training load and/or time under tension could be different; thus one group could have a more of a training stimulus yet both groups rated the training intensity the same.

This is obviously challenging when the training only included bodyweight training. Regardless, showing that both groups did equivalent training (as defined by volume and intensity) would strengthen the findings. One way to do this would be to present the sets and reps of the “regular conventional RG-specific warm-up, combining traditional flexibility, strength, and abdominal exercises, all aimed at RG body techniques (splits, bridges, leg kicks, body waves, v abs, candlesticks, feet work…etc.).” as a supplemental table. This would demonstrate the programs were similar in volume and intensity and then the lack of differences in RPE between groups would be evidence to strengthen the argument (instead of RPE being the sole evidence). I will leave this up to the authors and editors to decide if this an appropriate addition.

Author Response

[A] = answers from the authors.

Thank you again for your comments, all of them strengthen our work.

The methodological design was thought for both groups to train for the same amount of time so the volume would be the same for the EG and the CG. And then, since in gymnastics and this kind of activities is challenging to calculate the training intensity, calculating the RPE gave us a rate of perceived intensity, and this multiplied by the time of the session (RPE session) let us know an internal training load of each group to compare them.

Nevertheless, to affirm this answer “Since the RPE results did not show significant differences between the two groups in terms of load and volume, the EG is not doing more than the C.” we agree that a supplementary table containing the traditional rhythmic gymnastics warm-up exercises performed by the CG and showing the repetitions and sets, would help us demonstrate the load between groups had no significant differences.

Therefore, we added the supplementary material in the paper, and we attach it here as well.

Traditional rhythmic gymnastics warm-up for the control group

Content

Sets

Rep.

Time (min)

Different types of feet exercises on the floor with and without TheraBand

6-8

8

5-10

Leg flexibility movements for the side and the right and left front splits

12-15

1 x 8"

5-10

Body waves on the floor and standing up

4-6

8

5-10

Back flexibility (bridges, chest stands, back bends)

6-8

4 x 8"

5-10

Abdominals (V abs, curl abs, knee crunch, candlesticks)

4-6

8

5-10

Leg active flexibility (front, side, and back kicks)

6

8

5-10

We added the range of series that gymnasts did in the RG warm-up during the eight weeks of intervention.

This manuscript is a resubmission of an earlier submission. The following is a list of the peer review reports and author responses from that submission.

Round 1

Reviewer 1 Report

General comments:

Thank you for the opportunity to review the study. I commend the authors on conducting a challenging study. Interventions are difficult, especially in a younger age group. I also like the approach of CPT for use in this population of athletes. That said, I have some significant challenges with the manuscript.

First, the intro is too long with unnecessary details. It needs to be significantly shortened. Too much of reads as a literature review and some as a justification for certain dependent variables. Some of this then reappears in the methods.

Second, the control group training information is not provided and there is a lack of information on how the dependent variables were calculated. These should be included in the methods because without this information is impossible to fully evaluate the results.

Third, I question the use of the MANOVA. My concern is that many of the dependent variables are likely highly related and there will be issues of multicollinearity. I am not a statistician, but I believe there are issues with the sphericity assumption with repeated measures MANOVAs that must be accounted for. This, along with important details on the methods, makes it challenging to assess the results and the discussion.

Lastly, the discussion reads like a literature review without relating previous literature to the current results. There are some places where there is a good discussion of the results in relation to the literature, but there are other places where this could be improved (e.g., L427-473).  

Specific Comments

Intro

L43-44: Both of these sentences start with strength. Recommend rewording one of them.

L52-53: This sentence, specifically body difficulties is unclear.

L56-57: This sentence seems out of place and should come earlier.

L63-65: It is unclear to me how the flight phase depends on the quality of the push-off. Please explain what this means.

L68-72: Is the suggestion that isometric force is important for CS? If so, the link should be more explicitly made. 

L178-179: This sentence notes that jump testing occurred over 2 weeks, but the L175-176 states that testing occurred over 1 week intervals before and after the training. Please clarify.

L186: Please explain what is meant by higher loads than traditional training. Table 2 does not include loads.

Materials and Methods

L158-159: Please elaborate on what is meant by injuries that could disturb the testing or training.

L200-203: Was this for all exercises or just certain ones. Please clarify.

L204-205: How does RPE measure load? How was RPE used to modulate the training plan? More specifics are needed. I also do not see where RPE is used in the a

L222-223: This is unclear because it notes a professional RG coach developed the CPT under the supervision of professional RG coaches. This needs to be explained how one coach was different than other coaches.

L225-229: More details are needed here. It is challenging to compare to volume and intensity of the CG compared to the EG. I did not see this detail in the supplementary materials.

L243-244: Please clarify whether the 90 degree squat position was held before the jump or they were asked to descend to 90 degrees and then jump without a pause.

L255-256: Why was the jump with highest power value used vs. the jump with the highest jump height?

L264-266: It is unclear how jump height was calculated. Please elaborate on what is meant by the deformation of the plate’s sensors.

L270: Please explain what P3 means. It is also unclear how maximum slope was calculated.

Discussion

L416-418: This is unclear. Was the CPT training additional or different than the regular training. The way I read it was that it was not additional but different. Again, that is why it is important to provide the information about what the control group performed.

L481-485: This is all true, but again without an understanding of how RFD was calculated in this study it is difficult to assess the inclusion of this statement. I.e., it is unclear what portion of the force-time curve RFD was calculated from and how that relates to this reference.

L527-535: This might be true but considering both groups’ scores improved and only one group trained for physical components (e.g., explosive strength) and saw improvements in those components it likely means something else may explain the improvement. How do we know it was not just a result of more practice with the jumps over time, irrespective of the strength training program?

L554: This is unclear because there was a significant effect of time.

L561-562: Again this is unclear because there was an effect of time and the results state the follow-up tests were significant for both groups.

L579-580: I believe these references are inappropriate here. These are not experimental studies that provide evidence, but rather opinions of these authors. While I may personally agree with their perspectives, they do not support with evidence the contention of this sentence or the paragraph.

L614-615: It is unclear how RPE was used in this study (see previous comment) so this statement is unclear.

Table 1: Overall this is a good table. It would be helpful to include how often each block was performed each week. I’m assuming 3 times a week, but that isn’t clear.

Figures 3-6: I think these figures are challenging to interpret because of the scaling. For example, jump height was different on each of the jumps (either pre-post or pre-post and between group) but it is very hard to see the difference and it isn’t reported in the results. I think reconsidering how best to present the data is needed because the reader cannot fully evaluate the results.

Figure 7: The marks on the bars are unneeded and makes it confusing.

Author Response

Reviewer 1:

[R1] =

General comments:

Thank you for the opportunity to review the study. I commend the authors on conducting a challenging study. Interventions are difficult, especially in a younger age group. I also like the approach of CPT for use in this population of athletes. That said, I have some significant challenges with the manuscript.

[A] = Thank you, we appreciate your effort in reviewing the manuscript.

[R1] =  

First, the intro is too long with unnecessary details. It needs to be significantly shortened. Too much of reads as a literature review and some as a justification for certain dependent variables. Some of this then reappears in the methods.

[A] = The introduction has been shortened in order to avoid unnecessary details and repetitions. A full paragraph containing information about the CMJ and SLCMJ tests has been eliminated at this information is explained in the methodology section.

 [R1] =

Second, the control group training information is not provided and there is a lack of information on how the dependent variables were calculated. These should be included in the methods because without this information is impossible to fully evaluate the results.

[A] = Thank you, we have added some of the RG regular exercises underwent by the control group lines 296-297 as follows:

{...After that, the EG did the CPT program while the CG completed their regular conven-tional RG-specific warm-up, combining traditional flexibility, strength, and abdominal exercises, all aimed at RG body techniques (splits, bridges, leg kicks, body waves, v abs, candlesticks, feet work…etc.)....}

How dependent variables are calculated is provided in lines 347-356. The information was taken from the Kistler software manual.

{...The CMJ and SLCMJ variables measured were: Jump height of flight (Height), the height of the jump calculated from the deformation of the platform’s force sensors, measured in meters; vertical take-off velocity (Take-off), the velocity of the vertical movement at the time of take-off calculated from flight time, measured in m/s; the average power (AVE P) measured in watts (W); the average Force (AVE F) measured in newtons (N); the average velocity (AVE V) measured in m/s and the maximum concentric rate of force development (RFD) – P3, maximum slope of the force curve during the concentric phase of the jump measured in N/s. The concentric phase of the jump is the interval between the beginning of concentric movement of the muscule and the peak concentric force [55]....}

 [R1] =

Third, I question the use of the MANOVA. My concern is that many of the dependent variables are likely highly related and there will be issues of multicollinearity. I am not a statistician, but I believe there are issues with the sphericity assumption with repeated measures MANOVAs that must be accounted for. This, along with important details on the methods, makes it challenging to assess the results and the discussion.

[A] =The factors that we use in the study only have two levels: TIME (pre and post) and GROUP (control and experimental), this means that we cannot apply a sphericity test (for example the Mauchly test). To avoid problems with the differences in variances between the groups, the Levene test was previously applied, which reported that there was no difference in variances.

 [R1] =

Lastly, the discussion reads like a literature review without relating previous literature to the current results. There are some places where there is a good discussion of the results in relation to the literature, but there are other places where this could be improved (e.g., L427-473). 

 [A] = The discussion has been summarized so that only the comparisons with the results of our study appear and only give relevant information for the study.

Specific Comments

[R1] =

Intro

L43-44: Both of these sentences start with strength. Recommend rewording one of them.

[A] = Thank you, amended as follows

{...Strength is among the main physical capacities that have been found to contribute to performance in rhythmic gymnastics (RG) [1–4] and it is required for most of the movements and elements in this sport [5]...}

[R1] =

L52-53: This sentence, specifically body difficulties is unclear.

[A] = According to the rhythmic gymnastics code of points, body difficulties are the body elements that gymnasts perform to obtain part of the competition scores. There are 3 groups of body difficulties; jump/leap, balances and turns. This sentence pretends to explain that in RG competition routines the jump/leap elements predominate over the other two groups of body difficulties (or body elements).

 [R1] =

L56-57: This sentence seems out of place and should come earlier.

[A] = Amended as follows in lines 58-60:

{…The specific manifestations of strength in RG are strength-endurance and explosive strength [8,9]. Two studies compared the explosive strength of non-elite gymnasts to those of elite gymnasts [14,15], finding that the latter were more explosive jumpers. Gymnasts' lower limbs must exert force explosively to execute RG jump/leap skills [10,11]. RG demands a good deal of vertical jumping ability [12]...}

 [R1] =

L63-65: It is unclear to me how the flight phase depends on the quality of the push-off. Please explain what this means.

[A] = If the jump has a great push-off it will present more height, therefore the gymnasts will have more time to execute the jump/leap shape. This idea is explained in the text in lines 75-77:

{…The higher the athletes go during the jump, the greater their chances of succeeding in setting the body figure, and therefore the greater the likelihood of receiving good scores from the judges [9]...}

[R1] =

L68-72: Is the suggestion that isometric force is important for CS? If so, the link should be more explicitly made.

[A] = Thank you, we tried to clarify it as follows, lines 78-82:

{…In RG movements, isometric force plays a critical role in the pre-activation of the muscle before a contraction of the stretch-shortening cycle (SSC) [18]. Isometric strength exercises, like those used in core stability exercises (CS), are an innovative and highly effective way of increasing muscle mass, toning, and developing strength levels to improve fitness and athletic performance [19,20]...}

 [R1] =

L178-179: This sentence notes that jump testing occurred over 2 weeks, but the L175-176 states that testing occurred over 1 week intervals before and after the training. Please clarify.

[A] = Thank you, amended:

{All participants were tested two weeks before and two weeks after the eight-week intervention. Prior to the testing session, the order of the participants and of the two tests was randomly determined using a true random number generator to control for bias. Jump tests were carried out over two weeks in the pre-test and post-test by a sport-specialized physiotherapist, unaware of each subject’s allocation group.}

 [R1] =

L186: Please explain what is meant by higher loads than traditional training. Table 2 does not include loads.

[A] = Since the specific training included core exercises and plyometric exercises while the traditional training included specific RG warm-up, combining traditional flexibility, strength, and traditional abdominal exercises, the gymnasts in the experimental group will receive higher loads of core stability, core motor control and plyomteric exercises than the control group.

 [R1] =

Materials and Methods

L158-159: Please elaborate on what is meant by injuries that could disturb the testing or training.

[A] =   Injuries of any part of the body that wouldn’t allow the gymnasts to perform the specific training, the traditional training or the testing sessions. Injuries have a detrimental impact on  athletic success. Increased athletes availability improves chances of success. Injuries sustained both prior to and during competition increase risk of failure.

 [R1] =

L200-203: Was this for all exercises or just certain ones. Please clarify.

[A] = Thank you. Amended as follows:

{They were encouraged to hollow their navels while exhaling during the core stability exercises in each breath and each repetition, as repetitions were counted by breathing cycles.}

 [R1] =

L204-205: How does RPE measure load? How was RPE used to modulate the training plan? More specifics are needed. I also do not see where RPE is used in the a

[A] = The sentence of your review is not complete so we could not respond to the third comment.

Regarding the first and second questions, this text specifies how the RPE is measured and how is used to modulate the training plan. We added it as follows:

{Participants rated the intensity of the sessions through a rate of perceived exertion (RPE) session scale (sRPE), a valid method of quantifying exercise training during a wide variety of types of exercise [50]. A CPT trial was applied to check the gymnasts' perceived exercise intensity. The load to achieve a prescribed number of repetitions was adjusted to 7–8 values in the RPE scale (i.e., hard). Thirty minutes after every CPT session, all gymnasts (EG and CG) scored on the sRPE scale [50]. To obtain the sRPE the score was multiplied by the minutes of the session [51]. This outcome was used to modulate the training periodization plan. When values were lower than 7–8 sRPE a set was added in the exercises that were technically well executed. Similarly, the maintenance time of correctly executed isometric exercises was doubled in sRPE bellow 7–8.}

 [R1] =

L222-223: This is unclear because it notes a professional RG coach developed the CPT under the supervision of professional RG coaches. This needs to be explained how one coach was different than other coaches.

[A] = Yes, thank you, it is clarified as follows:

{An expert CS physiotherapist and a professional RG coach, both members of the research team developed the training protocol, the gymnasts underwent the CPT program under the supervision of professional RG coaches.}

 [R1] =

L225-229: More details are needed here. It is challenging to compare to volume and intensity of the CG compared to the EG. I did not see this detail in the supplementary materials.

[A] = Training intensity was calculated by the RPE session and no significant differences were found between groups, although, as we answered previously, since the control group is performing core stability and plyometric exercises 30min 3 times a week during 8 weeks they receive higher loads (more volume) of core stability and plyomteric exercises than the control group.

[R1] =

L243-244: Please clarify whether the 90 degree squat position was held before the jump or they were asked to descend to 90 degrees and then jump without a pause.

[A] = Thank you. Amended as follows:

{Participants were instructed to jump as fast and high as they could, first lowering their center of gravity and adopting a half squat position (i.e., descend the knees to 90º, feet shoulder-width apart), and then jumping pushing the ground vertically off the ground.}

 [R1] =

L255-256: Why was the jump with highest power value used vs. the jump with the highest jump height?

[A] = In studies based on an intervention for a certain time, it is better to use power, since it is a more stable variable. It may happen that during the intervention period the characteristics of the participants vary (for example, weight). Two jumps with the same height can be possible with different powers. If we compare the two groups based on the power variable, we do not depend on these variations and the results will be more consistent.

 [R1] = L264-266: It is unclear how jump height was calculated. Please elaborate on what is meant by the deformation of the plate’s sensors.

[A] = The force platform calculates the height of the jump from the takeoff speed. The platform software measures the deformation produced in the sensors and the speed at which it occurs, with this data it is able to report the height of each jump.

 [R1] = L270: Please explain what P3 means. It is also unclear how maximum slope was calculated.

[A] = P3 is the software designation for the portion of the force curve used to calculate the RFD. If the reviewer thinks it is confusing, it can be removed. in L 360-362 of the text it is explained how the RFD was calculated. The force platform software calculates the RFD from the maximum slope of the force curve from the start of the concentric movement to the force peak. The same software detects the beginning and the end by integrating the changes in the force curve.

 [R1] =

Discussion

L416-418: This is unclear. Was the CPT training additional or different than the regular training. The way I read it was that it was not additional but different. Again, that is why it is important to provide the information about what the control group performed.

[A] = The CPT training was added to the regular RG training. It formed a part of the training, including core stability and plyometric exercises different from which the gymnasts are used. Every RG training session was 3 hours long, and the CPT training lasted 30 min and was added after the general warm-up. While the EG performed the CPT program, the CG underwent their regular conventional RG-specific warm-up, combining traditional flexibility, strength, and abdominal exercises, all aimed at RG body techniques.

 [R1] =

L481-485: This is all true, but again without an understanding of how RFD was calculated in this study it is difficult to assess the inclusion of this statement. I.e., it is unclear what portion of the force-time curve RFD was calculated from and how that relates to this reference.

[A] = in L 360-362  of the text it is explained how the RFD was calculated. The force platform software calculates the RFD from the maximum slope of the force curve from the start of the concentric movement to the force peak. The same software detects the beginning and the end by integrating the changes in the force curve.

 [R1] =

L527-535: This might be true but considering both groups’ scores improved and only one group trained for physical components (e.g., explosive strength) and saw improvements in those components it likely means something else may explain the improvement. How do we know it was not just a result of more practice with the jumps over time, irrespective of the strength training program?

[A] = In this paragraph, and the following ones, we wanted to point out that there was no significant improvement in the judges' scores possibly because young gymnasts need more time to learn the leaps technique correctly and flexibility is a crucial factor for the judge's scores. Since the CPT didn't aim to improve flexibility nor it was focused on the leap technique, this might be the reason why the improvement was not reflected in the judge’s scores. Nevertheless, we suggest that when gymnasts learn the correct technique, counting on more explosive power in their lower limbs will help improve their leaps scores. In addition, a tendency of improvement in two of the three jumps evaluated was shown in the effect size results, specifically the ones that need higher lower limbs explosive strength to achieve the legs position (stag and split leap).

[R1] =

L554: This is unclear because there was a significant effect of time.

[A] = Thank you for your comment. We agree, since both groups improved the judge’s scores in the post-test, this explanation doesn’t apply. Therefore, the whole paragraph has been erased.

[R1] =

L561-562: Again this is unclear because there was an effect of time and the results state the follow-up tests were significant for both groups.

[A] = In this case, the effect size results show a tendency of improvement of the EG group over the CG on the split and the stag leap. This additional statistical analysis was made to find some possible differences between groups that might have goten unnoticed in the multivariate analysis.

 [R1] =

L579-580: I believe these references are inappropriate here. These are not experimental studies that provide evidence, but rather opinions of these authors. While I may personally agree with their perspectives, they do not support with evidence the contention of this sentence or the paragraph.

[A] = Thank you for your comment, we agree. Since the sentences in this paragraph might assume unclear statements, the content has been erased from the text.

 [R1] =

L614-615: It is unclear how RPE was used in this study (see previous comment) so this statement is unclear.

[A] = Answered in the previous comment.

 [R1] =

Table 1: Overall this is a good table. It would be helpful to include how often each block was performed each week. I’m assuming 3 times a week, but that isn’t clear.

[A] = Thank you, we suppose the comment was meant for table 2. Amended as follows in the table caption:

{DJ= drop jump, JDJ= jump-drop-jump, W= week, Block 1. circuit = mixed CS, balance, and plyometric exercises. Each week contained 3 CPT sessions. All exercises were performed with the right and left sides. All planks and lateral hip bridges were executed with passé, side leg, and arabesque positions and were maintained for 2 sec in each position. All DJ and JDJ were performed with a 30cm bench.}

 [R1] =

Figures 3-6: I think these figures are challenging to interpret because of the scaling. For example, jump height was different on each of the jumps (either pre-post or pre-post and between group) but it is very hard to see the difference and it isn’t reported in the results. I think reconsidering how best to present the data is needed because the reader cannot fully evaluate the results.

[A] = We have converted the information in the figures into a table for a better understanding of the data

 [R1] =

Figure 7: The marks on the bars are unneeded and makes it confusing.

[A] = We have converted the information in the figures into a table for a better understanding of the data

We greatly appreciate your comments on this manuscript as well as your time invested in analyzing our study

Reviewer 2 Report

The authors wrote an interesting manuscript comparing two training modalities (functional core and plyometric training and conventional training) on physical fitness and specific jump tests. The manuscript is well organized, and the information in each part is good.

Title

The article's title should be without a full stop.

Abstract

Line 32: "Results "should be bolted.

Introduction

The "Introduction" is relevant to the topic and generally provides a quality background for the research. The study's aim is clear; the authors want to determine the effect of two different training modalities on the physical fitness and specific jump tests of young rhythmic gymnasts.

Materials and Methods

Part "Materials and Methods" is mostly comprehensible but includes a few shortcomings.

Line 190: I do not think that, e.g., unilateral drop jumps are "low-impact plyometric exercises." Not for adults and definitely neither for 10-year-old girls.

Line 191: Jumps were performed from benches at 25 and 30 cm height. In line 199, the authors stated that all DJ and JDJ were performed with a 30 cm bench. Could the authors unify the height of the bench?

Line 226–228: What is the conventional RG-specific warm-up? Could the authors be more specific and describe mentioned exercises? This is the main point on which the study is built.

Line 311–312: A one-way repeated measure MANOVA was used for statistical analysis. In the abstract authors mentioned that a mixed model of ANOVA was used. Which statistical method was used?

Besides the assumption of normality for ANOVA also the assumption of homogeneity of variance needs to be checked. Moreover, there are other statistical assumptions associated with MANOVA.

In my opinion, the analysis of covariance (ANCOVA) should be used to assess the differences. It is helpful in pretest-posttest control group designs when the pretest is used as the covariate.

Results, Discussion, and Conclusions

"Results" are presented in a well-arranged way. The "Discussion" and "Conclusions" are conducted within the results' framework. I appreciate the study's limitations, where the authors know the limits and future research direction in this area. The gymnasts' ages ranged from 8 to 15 years, which is a very heterogeneous sample. Therefore, the results should be interpreted very carefully, and at least the statement that the results are not generalized for other age categories should be mentioned in the discussion. Moreover, it is tough to draw generally valid conclusions based on this.

The discussion part is too long and purposeless, so I recommend reducing this part – it may discourage potential readers from reading.

The same point about the references – too many references, this is not a systematic review.

Less is sometimes more.

In conclusion, the manuscript is well-written, and the authors use scientific language. Except for a few shortcomings, this is a quality paper.

On the other hand, I am looking for a rational reason to investigate 8- or 10-year-old-girls and do a specific training program. Girls in this age category (or coaches) should focus on other things like performance on a highly competitive level.

Author Response

Reviewer 2:

[R2] =

The authors wrote an interesting manuscript comparing two training modalities (functional core and plyometric training and conventional training) on physical fitness and specific jump tests. The manuscript is well organized, and the information in each part is good.

[A] = Thank you very much, we appreciate your effort in revising this manuscript.

[R2] =

Title

The article's title should be without a full stop.

[A] = Thank you, amended.

 [R2] =

Abstract

Line 32: "Results "should be bolted.

[A] = Thank you, amended.

 [R2] =

Introduction

The "Introduction" is relevant to the topic and generally provides a quality background for the research. The study's aim is clear; the authors want to determine the effect of two different training modalities on the physical fitness and specific jump tests of young rhythmic gymnasts.

[A] =  Thank you for your comment.

[R2] =

Materials and Methods

Part "Materials and Methods" is mostly comprehensible but includes a few shortcomings.

Line 190: I do not think that, e.g., unilateral drop jumps are "low-impact plyometric exercises." Not for adults and definitely neither for 10-year-old girls.

[A] = The sentence doesn’t indicate that unilateral drop jumps are low-impact jumps. It says: {In the plyometrics part of each training session, it is desirable to incorporate low-impact plyometric exercises (hops, jumps with two and one legs, and jumps from benches at a height of 25 and 30 cm) to acquire future adaptations and prevent injuries in children and adolescents [49].} These are guidelines from authors studying how to design resistance training. Responding to the next comment as well, these authors recommend drop jumps from 25-30cm benches and we opted for 30cm benches in our CPT training.

The DJ from our CPT training are all executed bilaterally except the DJ + scissors and the DJ + split leap since the gymnastics technique requires it.

[R2] =

Line 191: Jumps were performed from benches at 25 and 30 cm height. In line 199, the authors stated that all DJ and JDJ were performed with a 30 cm bench. Could the authors unify the height of the bench?

[A] = Thank you, answered above.

[R2] =

Line 226–228: What is the conventional RG-specific warm-up? Could the authors be more specific and describe mentioned exercises? This is the main point on which the study is built.

[A] =  Thank you, we added in the text the typical exercises performed in the RG-specific warm-up.

{The EG did the CPT program while the CG completed their regular conventional RG-specific warm-up, combining traditional flexibility, strength, and abdominal exercises, all aimed at RG body techniques (splits, bridges, leg kicks, body waves, v abs, candlesticks, feet work…etc.).}

[R2] =

Line 311–312: A one-way repeated measure MANOVA was used for statistical analysis. In the abstract authors mentioned that a mixed model of ANOVA was used. Which statistical method was used?

Besides the assumption of normality for ANOVA also the assumption of homogeneity of variance needs to be checked. Moreover, there are other statistical assumptions associated with MANOVA.

In my opinion, the analysis of covariance (ANCOVA) should be used to assess the differences. It is helpful in pretest-posttest control group designs when the pretest is used as the covariate.

[A] = The statistical analysis method used is the same. Some authors call it one-way repeated measures MANOVA and others call it mixed model ANOVA. We will unify the denomination with one-way repeated measures MANOVA.

Regarding the assumptions of the statistical model used, we must bear in mind that the factors we use in the study have two levels: TIME (pre and post) and GROUP (control and experimental), this means that we cannot apply a sphericity test. (for example the Mauchly test). To avoid problems with the differences in variances between the groups, the Levene test was previously applied, which reported that there was no difference in variances.

At first we also thought of including the pretest as a covariate, but the initial analyzes did not provide relevant information. Finally we decided on the current model.

[R2] =

Results, Discussion, and Conclusions

"Results" are presented in a well-arranged way. The "Discussion" and "Conclusions" are conducted within the results' framework. I appreciate the study's limitations, where the authors know the limits and future research direction in this area. The gymnasts' ages ranged from 8 to 15 years, which is a very heterogeneous sample. Therefore, the results should be interpreted very carefully, and at least the statement that the results are not generalized for other age categories should be mentioned in the discussion. Moreover, it is tough to draw generally valid conclusions based on this.

[A] =  Thank you, amended as follows in the study’s limitations paragraph:

{It is also important to point out that participants’ age range from 8 to 15 years, thus, results shouldn’t be generalized for other age categories.}

The discussion part is too long and purposeless, so I recommend reducing this part – it may discourage potential readers from reading.

The same point about the references – too many references, this is not a systematic review.

Less is sometimes more.

 [A] =  Thank you, we have reduced the discussion length and eliminated some references.

In conclusion, the manuscript is well-written, and the authors use scientific language. Except for a few shortcomings, this is a quality paper.

On the other hand, I am looking for a rational reason to investigate 8- or 10-year-old-girls and do a specific training program. Girls in this age category (or coaches) should focus on other things like performance on a highly competitive level.

[A] = We chose to investigate this age range since gymnasts start intensive training at a very young age. The CPT aimed to improve performance by improving the gymnasts' jumps execution and to investigate if training programs conducted successfully (such as core stability and plyometric training) in other sports would improve gymnasts' body skills. Certainly, an explosive strength enhancement of the lower limbs was achieved but an execution improvement is not so clear. It would be interesting to see if the same training conducted in elite gymnasts would enhance their leaps scores since they already have learned the correct technique.

Furthermore, in the majority of RG clubs, (at least in our region), gymnasts' age use to range from 6 to 15-year-old. Since we searched for gymnasts at the competitive level with at least one year experience, the sample was formed with 8-15-year-old gymnasts.

We greatly appreciate your comments on this manuscript as well as your time invested in analyzing our study

Reviewer 3 Report

I believe that the work entitled "A randomized clinical trial of the effects of an eight-week integrated functional core and plyometric training program on young rhythmic gymnasts' explosive strength" meets all the criteria of a scientific work. The research design, as well as the construction of the introduction, results and discussion meet the required form of scientific work. However, there are some issues that I would ask the authors for additional clarification.

Title: 

I would suggested authors to change title of paper. By my opinion It is to long and maybe to excluded "A randomized clinical trial“ from the title.  

Materials and Methods

Line 159-160

You mentioned that participants were randomly assigned either to the experimental group or control group using randomization software. Please can you be more specified about this software (version, manufacturer).

Line 240-244

In research, you mentioned that you used the CMJ test. Namely, in CMJ test starting position is upright position from which participant lowers the center of gravity of the body to a position of 90-degrees in the knee joint and immediately performs a vertical jump. 

In this case, where the starting position is with a 90-degree flexion in the knee joint, it implies the SQUAT JUMP test. 

Given that you tried to evaluate plyometric abilities with this test, a much more adequate test is the CMJ test, because it implies exactly eccentric-concentric contraction. Please clarify this part.

Also it is not clear based on figure 2. (a, b)

The discussion and conclusions of the paper are very well structured. The authors have studied and included findings from numerous previous studies that have studied a similar topic and have provided a quality explanation of the findings. Also, they clearly and systematically stated all the limitations of this study.

Author Response

Reviewer 3:

[R3] =

I believe that the work entitled "A randomized clinical trial of the effects of an eight-week integrated functional core and plyometric training program on young rhythmic gymnasts' explosive strength" meets all the criteria of a scientific work. The research design, as well as the construction of the introduction, results and discussion meet the required form of scientific work. However, there are some issues that I would ask the authors for additional clarification.

[A] = Thank you very much, we appreciate your effort in revising this manuscript.

[R3] =

Title:

I would suggested authors to change title of paper. By my opinion It is to long and maybe to excluded "A randomized clinical trial“ from the title. 

[A] = Thank you, we agree, ammended.

{The effects of an eight-week integrated functional core and plyometric training program on young rhythmic gymnasts’ ex-plosive strength}

[R3] =

Materials and Methods

Line 159-160

You mentioned that participants were randomly assigned either to the experimental group or control group using randomization software. Please can you be more specified about this software (version, manufacturer).

[A] = Thank you, amended as follows:

{Participants were randomly assigned either to the experimental group (EG, n=23) or the control group (CG, n=22) using the online randomization software

Research Randomizer (randomizer.org)}

[R3] =

  Line 240-244

In research, you mentioned that you used the CMJ test. Namely, in CMJ test starting position is upright position from which participant lowers the center of gravity of the body to a position of 90-degrees in the knee joint and immediately performs a vertical jump.

In this case, where the starting position is with a 90-degree flexion in the knee joint, it implies the SQUAT JUMP test.

Given that you tried to evaluate plyometric abilities with this test, a much more adequate test is the CMJ test, because it implies exactly eccentric-concentric contraction. Please clarify this part.

Also it is not clear based on figure 2. (a, b)

[A] = Amended as follows:

{2.6.1. Counter-movement jump (CMJ) test

The participants performed the CMJ from a standing position in accordance with Bosco’s [54] protocol. The gymnasts stood on the board barefoot and placed their hands on their hips to eliminate the influence of the potential impulse to move their arms. Participants were instructed to jump as fast and high as they could, first lowering their center of gravity and adopting a half squat position (i.e., descend the knees to 90º, feet shoulder-width apart), and then jumping pushing the ground vertically off the ground. They had to land with their ankles extended (toes pointed). Each subject completed three jumps, with one min of rest between them, and the jump with the highest power value was used for further analysis (Figure 2).

2.6.2. Single leg countermovement jump (SLCMJ) test

The participants performed SLCMJ from a standing position with one bare foot on the board, and the free lower limb bent and not touching the force plate. The gymnasts were instructed to place their hands on their hips and jump as fast and high as they could. The first movement consisted of a single leg half squat, i.e., knee at ~ 90º. Each participant completed six jumps (three with the right leg and three with the left as support), with one-minute rest periods between them. The jumps with the highest power value for the right and the left lower limbs were used for further analysis (Figure 2).}

A clarification has been also added in the figure caption:

{Figure 2. a) CMJ test and b) SLCMJ test. From a standing a position, gymnasts perform an SSC fast movement to jump.}

The discussion and conclusions of the paper are very well structured. The authors have studied and included findings from numerous previous studies that have studied a similar topic and have provided a quality explanation of the findings. Also, they clearly and systematically stated all the limitations of this study.

[A] = We greatly appreciate your comments on this manuscript as well as your time invested in analyzing our study

Round 2

Reviewer 1 Report

I appreciate the authors reworking the manuscript based on the suggestions. Many of the questions I had were addressed. However, there are still a few areas that are unanswered.

1.     I am still confused by the study procedures related to what each group did. One of the responses from the authors stated:

Training intensity was calculated by the RPE session and no significant differences were found between groups, although, as we answered previously, since the control group is performing core stability and plyometric exercises 30min 3 times a week during 8 weeks they receive higher loads (more volume) of core stability and plyomteric exercises than the control group.

But another response stated:

The CPT training was added to the regular RG training. It formed a part of the training, including core stability and plyometric exercises different from which the gymnasts are used. Every RG training session was 3 hours long, and the CPT training lasted 30 min and was added after the general warm-up. While the EG performed the CPT program, the CG underwent their regular conventional RG-specific warm-up, combining traditional flexibility, strength, and abdominal exercises, all aimed at RG body techniques.

And the Methods states the following:

During sessions, the EG and the CG did their general warm-up at the same time. This consisted of approximately 30 minutes of general activation and stretching exercises. After that, the EG did the CPT program while the CG completed their regular conventional RG- specific warm-up, combining traditional flexibility, strength, and abdominal exercises, all aimed at RG body techniques (splits, bridges, leg kicks, body waves, v abs, candlesticks, feet work...etc.). Afterward, the two groups reunited to undergo the rest of their regular training (the weekly plan is shown in supplementary materials).

So my question about the specific differences in volume and intensity between the 2 groups is unanswered. This information needs to be directly provided so the reader can understand if the differences were due to the inclusion of the CPT exercises or potentially due to differences in the EG group just doing more than the CG group.

2.     My concerns about the likelihood of a multicollinearity of the DVs in the MANOVA was not addressed. I think some of the DVs were highly correlated and this could impact the results.

3.     I asked if during the jump whether the participants were asked to pause at 90 degree and then jump or jump without a pause at 90 degrees. More information was added, but the specific question about pause or no pause was not included. I think this is important because a pause will impact the performance of the jump by diminishing the SSC.

A few other suggestions.

1.     I think I now understand that the authors used the Kistler software to derive the DVs versus calculating the DVs from the raw data. It would be significantly clearer to the reader if that was stated, such as all DVs were calculated by the XXX software.

2.     In the Methods jump height is described as being calculated from the deformation of the platforms force sensors. In the response the authors are more specific that takeoff speed is what was actually used to determine jump height. Adding in the specifics that takeoff velocity was what was used would be clearer to the reader. The authors added more information about the RFD calculation (i.e., RFD was calculated from start of concentric movement to the force peak) this is helpful and important. Although in the response they also added in the detail of “integrating the changes in the force curve”, which to me is confusing language. My suggestion is to add in more enough information about the DVs so the reader doesn’t have to go look in a software manual.

3.     I like the addition of Table 3. However, I am confused by the use of * and # without a legend explaining why. I think it is indicating a significant within and between subjects difference, respectively. If that is correct, I don’t think the symbols are needed since the p values are provided. I also suggest the p values have the same precision, i.e., same number of decimals places. Lastly, I suggest adding effect sizes. One, it would help the reader evaluate the differences. Two, it would be consistent with the way the technical scores are presented.

Author Response

[R1] = 

I appreciate the authors reworking the manuscript based on the suggestions. Many of the questions I had were addressed. However, there are still a few areas that are unanswered.

[A] = Thank you, we appreciate your effort in reviewing again the manuscript. 

[R1] =  

  1.     I am still confused by the study procedures related to what each group did. One of the responses from the authors stated:

Training intensity was calculated by the RPE session and no significant differences were found between groups, although, as we answered previously, since the control group is performing core stability and plyometric exercises 30min 3 times a week during 8 weeks they receive higher loads (more volume) of core stability and plyomteric exercises than the control group.

[A] = There is a mistake in this answer, we meant experimental group. 

But another response stated:

The CPT training was added to the regular RG training. It formed a part of the training, including core stability and plyometric exercises different from which the gymnasts are used. Every RG training session was 3 hours long, and the CPT training lasted 30 min and was added after the general warm-up. While the EG performed the CPT program, the CG underwent their regular conventional RG-specific warm-up, combining traditional flexibility, strength, and abdominal exercises, all aimed at RG body techniques.

And the Methods states the following:

During sessions, the EG and the CG did their general warm-up at the same time. This consisted of approximately 30 minutes of general activation and stretching exercises. After that, the EG did the CPT program while the CG completed their regular conventional RG- specific warm-up, combining traditional flexibility, strength, and abdominal exercises, all aimed at RG body techniques (splits, bridges, leg kicks, body waves, v abs, candlesticks, feet work...etc.). Afterward, the two groups reunited to undergo the rest of their regular training (the weekly plan is shown in supplementary materials).

So my question about the specific differences in volume and intensity between the 2 groups is unanswered. This information needs to be directly provided so the reader can understand if the differences were due to the inclusion of the CPT exercises or potentially due to differences in the EG group just doing more than the CG group

[A] = Thanks for your comment. Since the RPE results did not show significant differences between the two groups in terms of load and volume, the EG is not doing more than the CG. The difference is in the orientation of the CPT exercises. In all the EG exercises there was a high demand for core stabilization and, therefore, more isometric strength and control of the core muscles were required, as well as lower limbs explosive strength in the SSC exercises compared to the CG exercises performed in the RG traditional warm-up, which contained some traditional abdominals but also many exercises of flexibility and other RG techniques.

This information is clarified in the text lines 231-253:

{…Participants rated the intensity of the sessions throughout a rate of perceived exertion (RPE) session scale (sRPE), a valid method of quantifying exercise training during a wide variety of types of exercise [51]. A CPT trial was applied to check the gymnasts perceived exercise intensity. The load to achieve a prescribed number of repetitions was adjusted to 7–8 values in the RPE scale (i.e., hard). Thirty minutes after every CPT session, all gymnasts (EG and CG) scored on the sRPE scale [51]. To obtain the sRPE the score was multiplied by the minutes of the session [52]. There were no significant differences in sRPE between groups. This outcome was also used to modulate the CPT training periodization plan. When values were lower than 7–8 sRPE a set was added in the exercises that were technically well executed. Similarly, the maintenance time of correctly executed isometric exercises was doubled in sRPE bellow 7–8.

During sessions, the EG and the CG did their general warm-up at the same time. This consisted of approximately 30 minutes of general activation and stretching exercises. After that, the EG did the CPT program while the CG completed their regular conventional RG-specific warm-up, combining traditional flexibility, strength, and abdominal exercises, all aimed at RG body techniques (splits, bridges, leg kicks, body waves, v abs, candlesticks, feet work…etc.). Even though the CG also performed core exercises during the RG traditional warm-up, all the CPT exercises without exception contained a high demand for core stabilization, thus, more isometric strength and control of the core muscles were required. Furthermore, SSC exercises were not present in the CG RG warm-up, therefore, the EG received more specific loads of CS and SSC explosive strength exercises than the CG. Afterward, the two groups reunited to undergo the rest of their regular training (the weekly plan is shown in supplementary materials)...}

 [R1] = 

  1.     My concerns about the likelihood of a multicollinearity of the DVs in the MANOVA was not addressed. I think some of the DVs were highly correlated and this could impact the results.

[A] = We thought that multicollinearity would be a problem if we had used regression analysis. We have studied the subject and we have learned that a MANOVA can have problems when the dependent variables have a correlation coefficient above 0.9. We have checked the data between all the variables in the pre-post situations and in the three conditions (jump on two legs, right leg, left leg) and only twice is the 0.9 exceeded.

We have applied the Variance Inflation Factory, no variable exceeds 7.5.

 [R1] = 

  1.     I asked if during the jump whether the participants were asked to pause at 90 degree and then jump or jump without a pause at 90 degrees. More information was added, but the specific question about pause or no pause was not included. I think this is important because a pause will impact the performance of the jump by diminishing the SSC.

[A] =Thank you, amended in lines 264-272 for the CMJ and lines 274-281 for the SLCMJ.

[R1] = 

A few other suggestions.

  1.     I think I now understand that the authors used the Kistler software to derive the DVs versus calculating the DVs from the raw data. It would be significantly clearer to the reader if that was stated, such as all DVs were calculated by the XXX software.

 [A] = We have added this sentence to clarify that all dependent variables were calculated with the MARS software (Kistler) from the raw data. Lines 287-289.

{…Raw data were acquired (sampling rate 1,000 Hz) using the MARS software (Kistler, Winterthur, Switzerland) and all dependent variables were calculated from the mentioned software…}

Specific Comments

[R1] = 

  1.     In the Methods jump height is described as being calculated from the deformation of the platforms force sensors. In the response the authors are more specific that takeoff speed is what was actually used to determine jump height. Adding in the specifics that takeoff velocity was what was used would be clearer to the reader. The authors added more information about the RFD calculation (i.e., RFD was calculated from start of concentric movement to the force peak) this is helpful and important. Although in the response they also added in the detail of “integrating the changes in the force curve”, which to me is confusing language. My suggestion is to add in more enough information about the DVs so the reader doesn’t have to go look in a software manual.

[A] = Thank you, we have added information in the text so it is clearer for the reader. Amended in lines 291-301.

{...The CMJ and SLCMJ variables measured were: Jump height of flight (Height), the height of the jump calculated from the deformation of the platform’s force sensors, measured in meters; this parameter was calculated from the take-off speed. Vertical take-off velocity (Take-off), the velocity of the vertical movement at the time of take-off calculated from flight time, measured in m/s. The average power (AVE P) measured in watts (W). The average Force (AVE F) measured in newtons (N). The average velocity (AVE V) measured in m/s. The maximum concentric rate of force development (RFD) – P3 was also calculated. P3 is the software designation for the portion of the force curve used to calculate the RFD. The force platform software (MARS) calculates the RFD from the maximum slope of the force curve from the start of the concentric movement to the force peak....}

[R1] = 

  1.     I like the addition of Table 3. However, I am confused by the use of * and # without a legend explaining why. I think it is indicating a significant within and between subjects difference, respectively. If that is correct, I don’t think the symbols are needed since the p values are provided. I also suggest the p values have the same precision, i.e., same number of decimals places. Lastly, I suggest adding effect sizes. One, it would help the reader evaluate the differences. Two, it would be consistent with the way the technical scores are presented..

[A] = Thank you, since the p-value within and between subjects is specified in Table 3, the symbols * and # have been eliminated.

We greatly appreciate your comments on this manuscript as well as your time invested in analyzing our study in a 2nd round.